# Effect of an E-Prescription Intervention on the Adherence to Surgical Chemoprophylaxis Duration in Cardiac Surgery: A Single Centre Experience

**DOI:** 10.3390/antibiotics12071182

**Published:** 2023-07-13

**Authors:** Sofia Kostourou, Ilias Samiotis, Panagiotis Dedeilias, Christos Charitos, Vasileios Papastamopoulos, Dimitrios Mantas, Mina Psichogiou, Michael Samarkos

**Affiliations:** 1Infection Prevention Unit, Evaggelismos Hospital, 10676 Athens, Greece; kostour@med.uoa.gr (S.K.); vpapastamopoulos@evaggelismos-hosp.gr (V.P.); 2Department of Cardiac Thoracic and Vascular Surgery, Evaggelismos Hospital, 10676 Athens, Greece; isamiotis@evaggelismos-hosp.gr (I.S.); dedeilias@evaggelismos-hosp.gr (P.D.); chatitosch@otenet.gr (C.C.); 32nd Propaedeutic Department of Surgery, Laikon Hospital, Medical School, National and Kapodistrian University of Athens, 11527 Athens, Greece; dvmantas@med.uoa.gr; 41st Department of Medicine, Laikon Hospital, Medical School, National and Kapodistrian University of Athens, 11527 Athens, Greece; mpsichog@med.uoa.gr

**Keywords:** perioperative antimicrobial prophylaxis, surgical wound infection, cardiac surgical procedures, electronic prescribing

## Abstract

In our hospital, adherence to the guidelines for peri-operative antimicrobial prophylaxis (PAP) is suboptimal, with overly long courses being common. This practice does not offer any incremental benefit, and it only adds to the burden of antimicrobial consumption, promotes the emergence of antimicrobial resistance, and it is associated with adverse events. Our objective was to study the effect of an electronic reminder on the adherence to each element of PAP after cardiac surgery. We conducted a single center, before and after intervention, prospective cohort study from 1 June 2014 to 30 September 2017. The intervention consisted of a reminder of the hospital guidelines when ordering PAP through the hospital information system. The primary outcome was adherence to the suggested duration of PAP, while secondary outcomes included adherence to the other elements of PAP and incidence of surgical site infections (SSI). We have studied 1080 operations (400 pre-intervention and 680 post-intervention). Adherence to the appropriate duration of PAP increased significantly after the intervention [PRE 4.0% (16/399) vs. POST 15.4% (105/680), chi-square *p* < 0.001]; however, it remained inappropriately low. Factors associated with inappropriate duration of PAP were pre-operative hospitalization for <3 days, and duration of operation >4 h, while there were significant differences between the chief surgeons. Unexpectedly, the rate of SSIs increased significantly during the study (PRE 2.8% (11/400) vs. POST 5.9% (40/680), chi-square *p* < 0.019). The implemented intervention achieved a relative increase in adherence to the guideline-recommended PAP duration; however, adherence was still unacceptably low and further efforts to improve adherence are needed.

## 1. Introduction

Coronary artery by-pass grafting (CABG) and heart valve replacement (HVAR) result in significant benefits for the patients, in terms of years and Quality Adjusted Life Years (QUALIs), and might be more cost-effective than other interventions, e.g., percutaneous coronary angioplasty [1,2,3].

The incidence of surgical site infections following cardiac surgery (CS-SSI) ranges widely in the literature from 3.5 to 26.8% [4]. These infections include sternal wound infections (SWI) which are the most common, but also deep infections, i.e., mediastinitis or sternal osteomyelitis, which are a challenge to treat and may have dramatic consequences such as re-operation, long term antibiotic treatment or death [5]. The CS-SSIs associated mortality may reach 20% [4,6]. This is why, although technically both CABG and HVAR are generally classified as clean operations, peri-operative antimicrobial prophylaxis (PAP) is always recommended [7,8,9].

Except for the general risk factors for SSI, there are cardiac surgery-specific risk factors for SSI, the most important being the use of extracorporeal circulation, which affects the pharmacokinetics and pharmacodynamics of antimicrobials used for PAP [10,11,12]. The overall effectiveness of PAP has been established; however, it depends on the choice of the appropriate antimicrobial, and on the administration of the appropriate dose in the right time [13]. The chosen antimicrobial must be active against the usual pathogens on the site of surgery, i.e., staphylococci, which are involved in 50% of SSIs and Gram-negative pathogens, depending on the setting and risk factors [9]. The aim of PAP is to attain levels of antimicrobial above the minimal inhibitory concentration (MIC) for the duration of the operation. It is crucial that antimicrobial levels remain stable, therefore it is recommended that an intra-operative dose of PAP is given, when the duration of the procedure exceeds two half-lives of the antimicrobial [7,8,9]. Finally, duration of PAP is similarly important; in most cases, PAP consist of only one pre-operative dose. However, in cardiac surgery it has been suggested that PAP might be administered for up to 48 h, although there is no universal agreement on this [7,8,14,15]. It has been shown that prolongation of PAP beyond 48 h does not offer incremental benefits in SSIs prevention [7,8,16]. In contrast, increasing duration of PAP has been associated with adverse effects, such as development of acute kidney injury and *Clostridioides difficile* infection, in a duration-dependent fashion [17].

Furthermore, PAP accounts for a significant part of antimicrobial consumption in surgical departments, therefore inappropriately long duration of PAP would probably lead to unnecessary antimicrobial consumption and increase in hospital cost [18]. Attempts to improve adherence have been made in various centers and countries and been based on education and training, surveillance, and feedback, as well as in restrictive policies [19,20,21]. Data regarding adherence to PAP guidelines in Greece suggest that non-adherence is greater regarding PAP duration [22,23].

In our hospital, approximately 800 cardiac operations are performed each year. The Hospital Infection Unit, in collaboration with the Surgical Section of the hospital, has issued PAP guidelines, based on local microbiological data. In July 2015, an electronic reminder of the PAP Hospital Guidelines was incorporated into the drug prescription module of the hospital information system. The aim of the present study was to assess the effect of this intervention on the rates of adherence to PAP guidelines for cardiac surgery.

## 2. Results

### 2.1. Demographics and Comorbidities

Four hundred patients were included in the pre-interventions phase (PRΕ phase, 1 June 2014 to 30 June 2015) and 680 in the post-intervention phase (POST phase, 1 January 2016 to 30 September 2017). Most patients were male (74.7%, 807/1080), with a mean age of 65.8 years (SD = 11.0). Comorbidities were frequent as 22.3% (239/1074) patients were obese (BMI ≥ 30), 38.3% (414/1080) patients had diabetes, 25.1% (280/1080) had chronic kidney disease stage 3 or higher, and 33.1% (357/1080) patients had a history of smoking or were active smokers. The demographics and the comorbidities of patients in each phase are presented in Table 1.

Significant differences between the two phases of the study were the higher proportion of obese patients [67/394 (17.0%) vs. 174/680 (25.6%), Chi-square *p* = 0.001], of smoking [105/400 (26.3%) vs. 252/680 (37.1%), chi-square, *p* < 0.001], and of diabetes [133/400 (33.3%) vs. 281/680 (41.3%), chi-square *p* = 0.008] in the POST phase. There were no differences in other important variables.

### 2.2. Cardiac and Surgical History

There have been 1312 surgical procedures for 1080 patients, because a significant number of patients (213/1080, 19.7%) underwent combined procedures (e.g., CABG plus HVAR) during a single operative session (i.e., simultaneously via a common surgical incision). Of the remaining patients, 574/1080 (53.1%) underwent CAGB only, 224/1080 (20.7%) aortic valve procedures only and 69/1080 (6.4%) mitral valve procedures only. Most operative sessions were elective (751/1080, 69.5%). Pre-operative hospital stay was ≥3 days in 464/1080 (43.08%) patients and it was significantly shorter for immediate (salvage) operations (1.3 ± 2.8 days vs. 3.1 ± 3.2 days for elective procedures, *p* = 0.039). It was found that 42/1080 (3.9%) patients had a recent pre-operative infection, which was endocarditis in 26/1080 (2.4%) and respiratory tract infection in 11/1080 (1.0%). Information on the cardiac and surgical history is included in Appendix A. The only significant difference between the PRE and POST phase, regarding operation characteristics was that the median duration of the operative session was shorter in the POST phase (250 min vs. 230 min, Mann–Whitney, *p* < 0.001). The following variables were significantly more frequent in the PRE phase: Recent myocardial infarction [PRE 231/400 (57.8%) vs. POST 333/680 (49.0%), Chi-square *p* = 0.005], preoperative stay ≥3 days [ 196/400 (49.0%) vs. 268/680 (39.4%), Chi-square *p* = 0.002], and NYHA class III or IV [98/400 (24.5%) vs. 147/680 (21.6%) Chi-square *p* = 0.01] (Appendix A). The operative risk of the patients was assessed with multiple tools (ASA classification, NNISS, EuroSCORE II, and Cleveland Clinic score). There were no differences in any of the scores between PRE and POST patients, except for the ASA classification with a larger proportion of Class IV patients in the POST phase. Details are presented in Appendix A.

### 2.3. Adherence to PAP Duration

Adherence to the appropriate duration of PAP increased significantly after the intervention. However, it remained low [PRE 4.0% (16/399) vs. POST 15.4% (105/680), chi-square *p* < 0.001, OR = 4.37 95% CI 2.54–7.51]. Appropriateness of the duration of the first and second antimicrobial administered, when examined separately, also increased significantly (See Table 2).

When the adherence to the appropriate PAP duration by type of operation (CAGB, HVAR or composite operation) was examined, it was found that it increased significantly for isolated CABG and HVAR but not for composite operations (Table 2). Duration of PAP in days decreased significantly after the intervention (mean ± SD, PRE 5.07 ± 1.97 vs. POST 4.62 ± 2.51, Mann–Whitney test *p* < 0.001). The days on therapy (DOTs) per 100 patient days for all antimicrobials used in PAP also decreased significantly after the intervention (mean ± SD, PRE 91.83 ± 79.02 vs. POST 83.29 ± 41.75, Mann–Whitney test *p* < 0.007). To evaluate the effect of our intervention, we have also performed an ITS analysis, which showed a significant change in the slope after the implementation of the intervention (0.49 to 2.75, difference 2.26, 95% CI 0.49–4.036, Figure 1). Supremum Wald test confirmed the presence of a change point in the series (*p* = 0.02).

### 2.4. Factors Associated with Adherence to Appropriate Duration

The following variables were associated with adherence to PAP duration in univariate analysis: POST phase of the study (OR = 4.37, 95% CI 2.54–7.51), preoperative hospitalization ≥3 days (OR = 1.56, 95% CI 1.07–2.29), premature PAP initiation (OR = 0.45, 95% CI 0.23–0.85), duration of operation ≤4 h (OR = 3.12, 95% CI 2.02–4.83), and who the chief surgeon was (see below). Duration of operation (in minutes) was also associated with adherence to PAP duration (Mann–Whitney test, *p* < 0.001). Details of the univariate analysis are shown in Table 3 and Appendix A (for non-parametric results). In multivariate analysis with binary logistic regression, all the above variables were independently associated with appropriate PAP duration, except for premature PAP initiation (Table 4).

### 2.5. Individual Surgeon Adherence to PAP Duration

The data from five surgeons who had performed between 60 and 384 operations were analyzed. Three surgeons who had performed less than 20 operations were excluded from the analysis. The adherence to appropriate PAP duration per surgeon, before the intervention (“PRE”) ranged from 5.3% to 14.3% (chi square, *p* = ns). After the intervention (“POST”) adherence ranged from 2.2% to 26.8%, with a significant increase in two of the surgeons and non-significant changes in the remaining three (Appendix A). Interestingly, there were significant differences between chief surgeons and adherence to appropriate duration, when both phases of the study (PRE + POST) were analyzed together (chi square, *p* < 0.001).

### 2.6. Adherence to Other Individual Elements of PAP

The rate of appropriate choice of the first antimicrobial was high in the PRE phase, yet it increased significantly after the intervention (Table 3). Adherence to the choice of the second antimicrobial (Figure 2) was very low before the intervention and increased significantly after the intervention, but it was still low [PRE 4.5% (18/400) vs. POST 35.4% (241/680), chi-square *p* < 0.001]. It is noted that non-adherence both in the PRE and POST phase, mostly involved administration of combination PAP when it was not indicated (see Figure 2 for details). Adherence to the appropriate timing of the administration was very high for both antimicrobials before the intervention and it remained so and afterwards (Table 3). Intraoperative redosing was rarely needed for the first antimicrobial, which was vancomycin with its long half-life (0.9%, 10/1077), but it was indicated for the second antimicrobial in a significant proportion of operations (777/1034, 75.1%). Despite this, no intraoperative redosing was documented for any operation, i.e., adherence to redosing was 0% both before and after the intervention.

### 2.7. Secondary Outcomes of Antimicrobial Consumption

To investigate whether physicians used workarounds to prescribe extended PAP regimens (“stealth prescribing”), the number of patients which received antimicrobials during hospitalization for anything other than PAP indication was compared; the proportion did not differ between the two phases of the study: PRE 41.5% (166/400) vs. POST 39.4% (268/680), chi-square *p* = 0.49.

### 2.8. Other Secondary Outcomes

The incidence of SSI increased significantly after the intervention: PRE 2.8% (11/400) vs. POST 5.9% (40/680), chi-square *p* < 0.019. Most of the increase was due to an increase in deep SSIs [PRE 1.6% (6/400) vs. POST 4.3% (29/680)]. Documented post-operative infections other than SSIs also increased significantly [PRE 6.0% (24/400) vs. POST 10.3% (70/680), chi-square *p* < 0.016]. The most common types of post-operative other than SSI infections, were respiratory tract infections [PRE 1.6% (6/400); POST 3,5% 24/680)] and bloodstream infections [PRE 1.6% (6/400); POST 2,5% (17/680)]. Length of index hospitalization decreased marginally after the intervention [median, IQR 10 (9–13) vs. 10 (8–13), *p* = 0.001, while In-hospital mortality was not affected by the intervention [PRE 3.8% (15/400) vs. POST 4.8% (33/680), chi-square *p* = 0.39]. As expected, the length of hospitalization was significantly longer in patients with SSI [median, IQR 12 (10–18) vs. 10 (8–13), *p* = 0.0001]; however, In-hospital mortality did not differ between patients with and without SSI [3.9%, (2/51) vs. 4.5% (46/1029), *p* = 0.9].

## 3. Discussion

The aim of this study was to investigate the effect of a simple electronic reminder on the appropriateness of PAP, with focus on appropriate duration, which is probably the element of PAP with the lowest adherence [24,25]. The main finding of our study was that the intervention increased the proportion of patients which received appropriate duration of PAP from 4.0% to 15.4%. The improvement in appropriate duration of PAP was small, but it was confirmed using different methods (both PRE-POST comparison and ITS). In addition, other metrics of PAP duration such as mean duration in days and cumulative DOTs/100 patient days also improved. Duration of PAP in days decreased significantly after the intervention from 5.07 ± 1.97 to 4.62 ± 2.51 days. Similarly, the number DOTs/100 patient days for all antimicrobials used in PAP also decreased significantly after the intervention from 91.83 ± 79.02 to 83.29 ± 41.75.

Factors associated with adherence to appropriate PAP duration were preoperative hospitalization ≥3 days (OR = 1.87), duration of operation ≤4 h (OR = 2.32), and who the chief cardiac surgeon was. This suggests that both organizational (phase of the study, hospital stay) and procedural (operation duration, surgeon) factors can influence adherence to PAP duration. The association of longer preoperative hospitalization with better adherence cannot easily explained. One might argue that a short pre-operative hospitalization could be associated with immediate or salvage surgery, and in these severely ill patients, the surgeon might choose to prolong the duration of PAP. However, on the other hand, short pre-operative hospitalization also characterizes elective operations, in which there is no obvious reason for the prolongation of PAP. It is notable that the rate of preoperative stays of 3 days was higher in the PRE phase, suggesting possible improvements in pre-operative care in the POST phase. The association of longer operative times with lower adherence might be explained by their established association with a higher risk of SSI [26]. However, this strategy is not recommended, as there are no data suggesting that the extension of PAP reduces the risk of SSI [27]. It is important to note that the duration of the operative session was shorter in the POST phase (Appendix A), which might be indicative of improvements in surgical efficiency or alterations in surgical techniques over time. The differences in adherence to PAP duration between cardiac surgeons were significant; after the intervention, adherence improved significantly in two out of five cardiac surgeons, it improved but was not statistically significantly in one, and it deteriorated, albeit non-significantly, in the last one (Appendix A). This finding has important implications, as it underlines the differences in individual responses to the same intervention and the need to “personalize” the approach to implementation of the intervention [28]. The causes of the different responses of individual healthcare workers could not be addressed by our study, and qualitative research is needed for this issue.

Among the different forms of non-adherence to PAP guidelines, inappropriate duration seems to be the most important, since prolonged duration has the largest impact on excessive antimicrobial consumption. Even if all other elements of PAP are appropriate, inappropriate duration leads to inappropriate use of antimicrobials. In addition, increasing duration of antimicrobial prophylaxis was associated with higher odds of acute kidney injury and C difficile infection [17].

The importance of appropriate duration has been highlighted in the second European point prevalence survey, where 14.2% of antimicrobial agents in hospitals were prescribed for PAP and 54.3% of PAP courses were prescribed for more than 1 day [29,30]. In Greece, the problem is more pressing as PAP accounts for 24.2% of antimicrobial courses, with almost 75% of patients receiving PAP for >1 day. The country ranks fifth in Europe in this metric. These results cannot compare directly to our data, since we have studied cardiac operations only, which is the type of operation in which PAP is allowed for up to 48 h [8]. Chorafa et al. recently published a study of a multimodal intervention in five hospitals in Greece, appropriate PAP duration increased from 33.4% to 60.3% [23]. In this study, cardiac surgery accounted for approximately 31% of the operations; however, the large discrepancy in improvement between that study and the current cannot be attributed to this only. The fact that the pre-intervention adherence in our study was much lower in comparison to the study of Chorafa et al. suggests that local factors have played a role in low adherence. A possible reason for the smaller improvement in the current study is the lack of a multimodal approach, as our intervention consisted of a simple reminder during prescription, in contrast to the study of Chorafa et al. which used a multimodal approach in the context of a national initiative. The effectiveness of multimodal antimicrobial stewardship interventions is well established; however, these interventions require infrastructure and are costly. Thus, it might be difficult to employ them routinely at hospital level (i.e., outside wider initiatives) and over long periods [31]. Simple interventions, such as the one studied, have smaller benefits, but they are easier to establish. Furthermore, they might be useful as a first step in changing the healthcare workers culture and establishing a more comprehensive antimicrobial stewardship program or as a complement to multimodal interventions [32].

Regarding the adherence to guidelines of the other elements of PAP, appropriate choice of first antimicrobial and adherence to the appropriate timing of the administration were already high in the PRE phase, and either increased further or remained high. However, the appropriate use of a second antimicrobial was very low before the intervention and although it increased after the intervention, it remained relatively low. Non-adherence to the choice of the second antimicrobial involved administration of a second antimicrobial even when it was not indicated. According to the hospital guidelines, a second antimicrobial was indicated for patients with a pre-operative hospitalization lasting at least 3 days. However, most patients with pre-operative hospitalization for <3 days, received two antimicrobial drugs as well. The existing guidelines (i.e., the joint guidelines of American Society of Health-System Pharmacists, the Infectious Diseases Society of America, the Surgical Infection Society, and the Society for Healthcare Epidemiology of America and the guidelines of the Society of Thoracic Surgeons) recommend a beta-lactam (usually a first- or second-generation cephalosporin) as the preferred agent, while vancomycin is only recommended when there is allergy to beta-lactams, or there is a concern regarding the high prevalence of methicillin-resistant *Staphylococcus aureus* (MRSA) infections [7,8,9]. Vancomycin lacks any activity against Gram-negative pathogens, and it may even be less effective than beta-lactams when the pathogen is susceptible. Thus, the guidelines suggest that an agent active against Gram-negative pathogens be considered along vancomycin, especially when the hospital reports deep SSIs or bloodstream infections after cardiac surgery from Gram-negative pathogens [9]. In our setting, there was both a high prevalence of MRSA infections and widespread Gram-negative multi-drug resistance pathogens. Therefore, the Hospital Infection Unit decided to recommend vancomycin as a first choice and the addition of an agent active against resistant Gram-negative pathogens, such as piperacillin/tazobactam or meropenem for the higher risk patients, i.e., those with pre-operative hospitalization for ≥3 days. Finally, although intraoperative dosing was needed in a large proportion of patients which received a second antimicrobial, it was never administered. These issues are probably the result of defective knowledge regarding the hospital guidelines, and could have been avoided with a multimodal approach, which would have included educational interventions.

An unexpected result was the significant increase in SSIs after the intervention. Although the incidence of SSIs increased it remained within acceptable levels. The role of the intervention in this increase is difficult to estimate, as in logistic regression analysis (Appendix A), SSIs overall were associated with several other factors such as diabetes mellitus, female sex, and history of neoplasm.

Another finding worth mentioning is a significant increase in the POST phase of patients with obesity, diabetes, or history of smoking. This might suggest a trend which requires increased attention for better management of these comorbidities.

Our study had several advantages, since a large number of operations was studied prospectively, over an extended period. Since the focus was on cardiac surgery, the findings are easier to interpret and can be applied to this type of surgery. Our findings are robust since more than one method and metric were used to confirm the benefit of the intervention regarding the primary outcome.

A disadvantage is that the setting of the study is characterized by high levels of resistance, which is reflected in the recommendations of the hospital guidelines. Thus, our results might be applicable only in such settings.

## 4. Materials and Methods

### 4.1. Study Design and Patient Population

This was a single center, prospective interventional before–after study, conducted at Evaggelismos Hospital, a 945 bed, tertiary hospital. Each year, approximately 9500 surgical operations are performed in the hospital, approximately 800 of which are cardiac. There were two phases in the study, a pre-intervention phase (PRE, 1 June 2014 to 30 June 2015) and a post-intervention phase (POST, 1 January 2016 to 30 September 2017). The intervention was implemented on 1 July 2015, and a run-in period of six months was allowed (1 July 2015 to 31 December 2015). All coronary artery by-pass grafting (CABG), heart valve replacement (HVAR), or composite cardiac operations which were performed every other month of the study period were included in the study. Thus, data were collected for 7 months in the PRE phase and for 11 months in the POST phase. Only the initial operation for each patient during the study period was included. The study protocol was approved by the Institutional Review Board of the hospital (No. 73/07-04-2014).

### 4.2. Intervention

The intervention consisted of an extra step on the antimicrobial prescribing through the hospital information system. The prescribing physician initially had to select whether the antimicrobial would be therapeutic or prophylactic. When the physician selected “prophylactic” a reminder with a link to the PAP Hospital Guidelines appeared. The intervention was not restrictive regarding the choice of antibiotic; however, the amount of antimicrobial dispensed was adjusted to the appropriate duration, i.e., 48 h for cardiac surgery. The intervention was presented to the medical and nursing staff of the Cardiac Surgery Department and of the dedicated Cardiac and Thoracic Surgery ICU of the hospital before the run-in period, along with training sessions regarding PAP.

### 4.3. Data Collection

Data regarding patient demographics, medical history, PAP, operation, and outcomes were collected. Surgical urgency was defined according to the NCEPOD [33]. Physical patient medical records, as well as the laboratory and the pharmacy module of the hospital information system were used. Scores regarding risk assessment of heart related morbidity and mortality in patients undergoing cardiac surgery, i.e., American Society of Anesthesiologists (ASA) score, National Nosocomial Infections Surveillance System (NNIS) score, Euroscore II, and Cleveland Score were calculated [34,35,36].

### 4.4. Outcomes

The primary outcome of the study was adherence to the guideline-suggested duration of PAP (i.e., the proportion of patients who received PAP for up to 48 h). Secondary outcomes included duration of PAP in days and in days on therapy (DOTs), adherence to the other elements of PAP (i.e., choice of appropriate antibiotic, timing of initiation of PAP, intra-operative repeat dosing), postoperative use of antibiotics during hospitalization, the incidence rate of SSI, length of index hospitalization, and in-hospital mortality.

The appropriateness of PAP was assessed in terms of adherence to the hospital guidelines for each element of PAP. The institutional guidelines for heart surgery recommend the administration of vancomycin alone (15 mg/kg) for patients with pre-operative hospitalization of less than 3 days, or alternatively teicoplanin (6 mg/kg rounded to the nearest 200 mg) while for patients with a pre-operative hospitalization of at least 3 days, the administration of a combination of vancomycin 15 mg/Kg and either piperacillin/tazobactam 4.5 g or meropenem 1 g is recommended. The administration of a second antimicrobial was considered as appropriate when both conditions were met: there was indication for combination PAP, and the correct antimicrobial was selected. In any other case, the administration was considered inappropriate (i.e., administration of any second antimicrobial when combination PAP was not indicated, not administering a second antimicrobial when it was indicated, and administration of combination PAP but with a second antimicrobial not included in the institutional guidelines.)

The timing was considered appropriate if the infusion of antimicrobials had been completed within 60 min of the surgical incision. Institutional PAP Guidelines suggested the administration of intra-operative repeat dosing in accordance with the joint guidelines [7]. Redosing was considered appropriate if it was both indicated and administered using the appropriate antimicrobial. A duration of PAP less than 48 h from the operation was considered appropriate. When a combination of antimicrobials was prescribed, appropriate duration was assessed separately for each one. Administration of antimicrobials before the day of the surgery was considered “unjustified” if there was no clinically or microbiologically documented infection. All decisions regarding appropriateness were made jointly by the Infection Control nurse and an Infectious Diseases physician. Except for PAP no other pharmacological intervention for SSI prevention was used, e.g., gentamicin-impregnated sponges.

Surveillance for post-operative infections was performed using culture results of all biological samples and by medical record review of included patients. Bacteraemias, respiratory tract infections, urinary tract infections, C. difficile infections, and SSIs were monitored according to the Centers for Disease Control and Prevention definitions [37].

### 4.5. Statistical Analysis

Qualitative variables were described as absolute (n) and relative (%) frequencies. For comparison of proportions Pearson’s chi square or Fisher’s exact test was used, as appropriate. Quantitative variables were described using mean and standard deviation or median and interquartile range, depending on the normality of distribution. Normality was assessed using the Kolmogorov–Smirnov test and via visual inspection of histograms. Quantitative variables were compared using Student’s *t* test or Mann–Whitney test, as appropriate. Variables associated with specific outcomes in univariate analyses were included in logistic regression models, to explore the independent risk factors of these outcomes. For the evaluation of the primary outcome, we additionally performed an interrupted time series (ITS) analysis. All comparisons were two-sided, and the level of statistical significance was set at *p* < 0.05. For the statistical analysis IBM SPSS Statistics^®^ version 26 was used, except for the ITS analysis for which we have used the RITS toolbox [38].

## 5. Conclusions

In this study, we have found that a simple intervention in the electronic prescription through the hospital information system of antimicrobials for PAP, was associated with a small but significant increase in the adherence to the appropriate PAP duration. This was confirmed by ITS analysis and by other metrics such as average duration of PAP. The effect of our intervention was small in comparison with the effect of multimodal strategies, and apparently further efforts are needed to improve adherence; however, we suggest that such small interventions are useful as a complement to other, larger scale, interventions and have the advantage of easy implementation and possibly better sustainability.

## Figures and Tables

**Figure 1 antibiotics-12-01182-f001:**
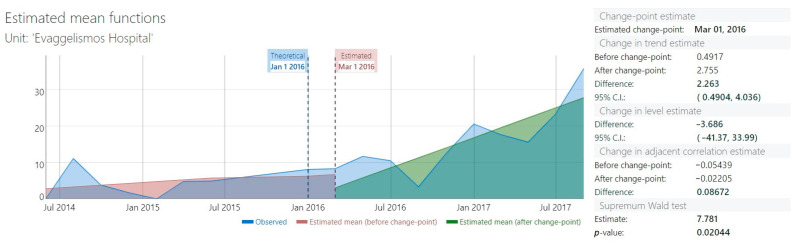
Interrupted time series analysis of PAP duration. The intervention was fully implemented in January 2016.

**Figure 2 antibiotics-12-01182-f002:**
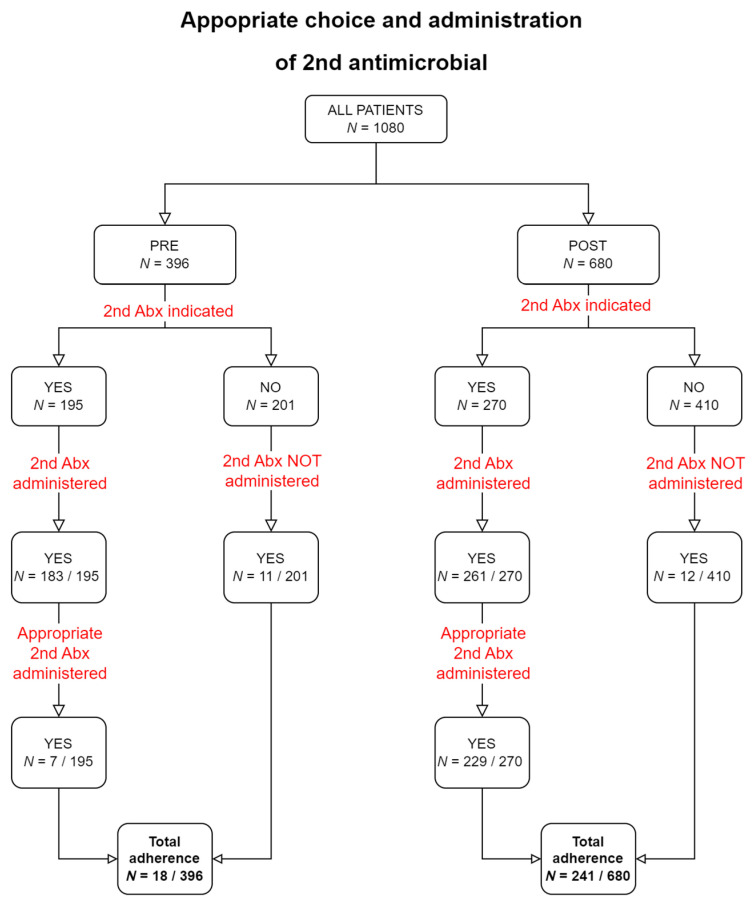
Flowchart of the appropriateness of the choice of the 2nd antimicrobial.

**Table 1 antibiotics-12-01182-t001:** Patient demographics and comorbidities.

	Pre-Intervention, N (%)	Post-Intervention, N (%)	Total, N	*p* Value
Gender (Male)	290/400 (72.5%)	517/680 (76.0%)	807/1080 (74.7%)	0.197
Mean Age (years, (SD)	66.2 (10.7)	65.7 (11.2)	65.8 (11.0)	0.53
ΒΜΙ ≥ 30 Kg/m^2^	65/1080 (16.5%)	174/1080 (25.6%)	239/1080 (22.3%)	0.02
Mean BMI, Kg/m^2^ (SD)	27.0 (3.9)	27.5 (4.3)	27.3 (4.2)	0.083
History of smoking	105/399 (26.3%)	252/680 (37.1%)	357/1079 (33.1%)	<0.001
Diabetes	133/400 (33.3%)	281/680 (41.3%)	414/1080 (38.3%)	0.008
Hypertension	368/400 (92.0%)	638/680 (93.8%)	1006/1080 (93.1%)	0.252
Dyslipidemia	287/399 (71.9%)	495/680 (72.8%)	782/1079 (72.5%)	0.759
Immunosuppression (therapy/disease)	25/400 (6.3%)	51/680 (7.5%)	76/1080 (7.0%)	0.438
Peripheral vascular disease	42/400 (10.5%)	68/679 (10.0%)	110/1079 (10.2%)	0.799
Chronic kidney disease (G3 or higher)	123/400 (30.7%)	157/680 (23.1%)	280/1080 (25.1%)	0.515
Chronic lung disease	24/400 (6.0%)	57/680 (8.4%)	81/1080 (7.5%)	0.151
History of Cerebrovascular disease	12/400 (3.0%)	34/680 (5.0%)	46/1080 (4.3%)	0.116
History of malignancy	17/400 (4.3%)	32/680 (4.7%)	49/1080 (4.5%	0.728
Thyroid disease	35/400 (8.8%)	70/679 (10.3%)	105/1079 (9.7%)	0.404
Other comorbidities	85/400 (21.3%)	185/680 (27.2%)	270/1080 (25.0%)	0.029

**Table 2 antibiotics-12-01182-t002:** Adherence to individual PAP elements.

PAP Element	Pre-Intervention, N (%)	Post-Intervention, N (%)	*p* Value
**Appropriate duration**			
Appropriate duration 1st antibiotic	19/399 (4.8%)	116/680 (17.1%)	<0.001
Appropriate duration 2nd antibiotic	59/375 (15.7%)	163/659 (24.7%)	0.001
Appropriate duration overall	16/399 (4.0%)	105/680 (15.4%)	<0.001
Appropriate duration overall—CABG only	11/229 (4.8%)	58/345 (16.8%)	<0.001
Appropriate duration overall—HVAR only	2/109 (1.8%)	34/216 (15.7%)	<0.001
Appropriate duration overall—Composite operation	3/61 (4.9%)	13/119 (10.9%)	0.018
**Appropriate choice**			
Appropriate choice 1st antibiotic	381/400 (95.3%)	673/680 (99.0%)	<0.001
Appropriate choice 2nd antibiotic	18/396 (4.5%)	241/680 (35.4%)	<0.001
Appropriate choice of antibiotic overall	16/400 (4.0%)	238/680 (35.0%)	<0.001
**Appropriate timing of initiation**			
Appropriate timing 1st antibiotic	396/400 (99.0%)	679/680 (99.9%)	0.046
Appropriate timing 2nd antibiotic	350/400 (92.8%)	625/680 (94.8%)	0.19 (ns)
Appropriate timing Overall	373/400 (93.3%)	645/680 (94.9%)	0.27 (ns)
**Appropriate redosing**			
Appropriate redosing 1st antibiotic	0/4 (0%)	0/6 (0%)	n/a
Appropriate redosing 2nd antibiotic	0/183(0%)	0/194 (0%)	n/a

**Table 3 antibiotics-12-01182-t003:** (**A**) Univariate analysis of qualitative variables possibly associated with adherence to PAP duration: Demographics, history, pre-operative factors. (**B**) Univariate analysis of qualitative variables possibly associated with adherence to PAP duration: Risk scores and operative factors.

(**A**)								
				**Appropriate PAP Duration**				
**Variable**		**Category**		**No (n, %)**		**Yes (n, %)**	**OR (95% CI)**	** *p* **
Phase of the study		PRE		383 (96.0%)		16 (4.0%)		
		POST		575 (84.6%)		105 (15.4%)	4.37 (2.54–7.51)	<0.001
Sex		Male		710 (88.0%)		97 (12.0%)		
		Female		248 (91.2%)		24 (8.8%)	0.70 (0.44–1.13)	0.149
Age > 65		>65		532 (89.4%)		63 (10.6%)		
		≤65		426 (88.0%)		58 (12.0%)	0.87 (0.59–1.27)	0.47
BMI (cat)		Normal		293 (89.3%)		35 (10.7%)		
		Overweight		444 (87.9%)		61 (12.1%)		
		Obese		214 (89.5%)		25 (10.5%)	n/a	0.739
Diabetes mellitus		Yes		361 (87.2%)		53 (12.8%)		
		No		597 (89.8%)		68 (10.2%)	1.29 (0.88–1.89)	0.192
CKD		Yes		26 (78.8%)		7 (21.2%)		
		No		932 (89.1%)		114 (10.9%)	2.20 (0.93–5.18)	0.65
COPD		Yes		75 (92.6%)		6 (7.4%)		
		No		883 (88.5%)		115 (11.5%)	0.61 (0.262–1.44)	0.259
History of neoplasm		Yes		40 (81.6%)		9 (18.4%)		
		No		918 (89.1%)		112 (10.9%)	1.84 (0.87–3.90)	0.104
Immunosuppression		Yes		65 (85.5%)		11 (14.5%)		
		No		893 (89.0%)		110 (11.0%)	1.37 (0.70–2.68)	0.35
Permanent pacemaker (preop)		Yes		29 (82.9%)		6 (17.1%)		
		No		929 (89.0%)		115 (11.0%)	1.67 (0.68–4.11)	0.258
History of prior cardiac surgery		Yes		165 (89.7%)		19 (10.3%)		
		No		793 (88.6%)		102 (11.4%)	0.89 (0.53–1.50)	0.675
History of endocarditis		Yes		27 (93.1%)		2 (6.9%)		
		No		930 (88.7%)		119 (11.3%)	0.58 (0.14–2.46)	0.454
Preoperative hospitalization ≥3 days		Yes		400 (86.2%)		64 (13.8%)		
		No		558 (90.7%)		57 (9.3%)	1.56 (1.07–2.29)	0.020
Preoperative infection		Yes		37 (88.1%)		5 (11.9%)		
		No		921 (88.8%)		116 (11.2%)	1.07 (0.41–2.78)	0.885
Preoperative CRP		>0.5 mg/dL		294 (88.3%)		39 (11.7%)		
		<0.5 mg/dL		664 (89.0%)		82 (11.0%)	1.07 (0.72–1.61)	0.729
Preoperative WBC		Low		43 (93.5%)		3 (6.5%)		
		Normal		822 (88.8%)		104 (11.2%)		
		High		93 (86.9%)		14 (13.1%)	n/a	0.498
Critical preoperative status		Yes		23 (92.0%)		2 (8.0%)		
		No		933 (88.7%)		119 (11.3%)	0.68 (0.16–2.93)	0.604
Premature PAP initiation		Yes		174 (94.1%)		11 (5.9%)		
		No		784 (87.7%)		110 (12.3%)	0.45 (0.24–0.86)	0.013
(**B**)								
						**Appropriate PAP Duration**		
**Variable**			**Category**			**No (n, %)**	**Yes (n, %)**	** *p* **
Type of procedure			CABG only			505 (88.0%)	69 (12.0%)	
			HVAR only *			261 (89.4%)	31 (10.6%)	
			Composite			192 (90.1)	21 (9.9%)	0.646
Surgical urgency			Immediate			6 (100.0%)	0 (0.0%)	
			Emergent			10 (100.0%)	0 (0.0%)	
			Expedited			272 (87.2%)	40 (12.8%)	
			Elective			670 (89.2%)	81 (10.8%)	0.397
ASA score			ΙΙ			241 (90.9%)	24 (9.1%)	
			ΙΙΙ			479 (88.4%)	63 (11.6%)	
			IV			237 (87.5%)	34 (12.5%)	0.404
BASIC RISK INDEX			0			4 (100.0%)	1 (0.0%)	
			1			753 (87.7%)	106 (12.3%)	
			2			190 (93.6%)	13 (6.4%)	
			3			10 (83.3%)	2 (16.7%)	0.083
EUROSCORE category			Low			596 (89.0%)	74 (11.0%)	
			Intermediate			258 (86.6%)	40 (13.4%)	
			High			91 (93.8%)	6 (6.2%)	
			Very high			13 (92.9%)	1 (7.1%)	0.243
CLEVELAND category			Low			527 (88.6%)	68 (11.4%)	
			Intermediate			303 (88.9%)	38 (11.1%)	
			High			127 (89.4%)	15 (10.6%)	0.956
Surgeon			A			158 (79.4%)	41 (20.6%)	
			B			57 (95.0%)	3 (5.0%)	
			C			147 (87.5%)	21 (12.5%)	
			D			339 (88.5%)	44 (11.5%)	
			E			238 (96.4%)	9 (3.6%)	<0.001
Duration of operation ≤4 h			≤4 h			481 (83.9%)	92 (16.1%)	
			>4 h			474 (94.2%)	29 (5.8%)	<0.001

* Data missing for one patient.

**Table 4 antibiotics-12-01182-t004:** Binary logistic regression for appropriate PAP duration.

Variable	Odds Ratio	Lower 95% C.I.	Upper 95% C.I.	*p*
Phase of the study	4.708	2.545	8.71	0.000
Sex	0.655	0.395	1.089	0.099
Diabetes mellitus	1.224	0.807	1.855	0.303
History of endocarditis	0.329	0.043	2.498	0.304
History of neoplasm	1.992	0.689	5.757	0.171
Immunosuppression	0.995	0.388	2.550	0.843
Permanent pacemaker (preop)	2.097	0.791	5.559	0.127
Premature PAP initiation	0.769	0.337	1.753	0.490
Prior cardiac surgery	0.864	0.494	1.510	0.515
Preoperative infection	2.373	0.530	10.618	0.283
Preoperative hospitalization ≥3 days	1.874	1.235	2.843	0.003
Critical preoperative status	0.799	0.164	3.890	0.693
Duration of operation >4 h (1)	2.328	1.425	3.805	0.003
Surgeon				0.001
Surgeon A *	1.492	0.908	2.452	0.829
Surgeon B	0.342	0.100	1.168	0.007
Surgeon C	0.874	0.484	1.58	0.000
Surgeon E	0.302	0.140	0.655	0.010

* Comparison with surgeon with highest number of procedures (Surgeon D).

## Data Availability

The data presented in this study are available on request from the corresponding author. The data are not publicly available due to privacy restrictions.

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
