# Peer review of "Effect of an E-Prescription Intervention on the Adherence to Surgical Chemoprophylaxis Duration in Cardiac Surgery: A Single Centre Experience"

_antibiotics, 2023, doi:10.3390/antibiotics12071182_

Round 1

Reviewer 1 Report

Overall, this study seems to demonstrate the complex interplay between patient demographics, comorbidities, procedural factors, and adherence to medical guidelines. Further efforts could focus on improving adherence rates to PAP guidelines, potentially considering individual patient and procedural characteristics. Additionally, rising comorbidity rates in the POST phase point to the need for more robust preventive measures and management strategies for these conditions.

Based on the data provided by the authors, here are some comments and possible interpretations:

1: Demographics and Comorbidities: Most patients in this study were male and had a mean age of 65.8 years. Comorbidities were common, with obesity, diabetes, chronic kidney disease stage 3 or higher, and a history of smoking being the most prevalent. A significant increase was observed in the POST phase for the proportion of patients who were obese, were smokers, and had diabetes. This might suggest a trend of rising prevalence for these conditions, warranting heightened attention for better management of these comorbidities.

2: Cardiac and Surgical History: Many patients underwent combined procedures during a single operative session. The most common procedure was CABG (coronary artery bypass grafting), and most procedures were elective. Notably, the duration of the operative session was shorter in the POST phase, which might be indicative of improvements in surgical efficiency or alterations in surgical techniques over time. Also, the rates of recent myocardial infarction and preoperative stays of 3 days or more were higher in the PRE phase, suggesting possible improvements in pre-operative care or patient health status in the POST phase.

3: Adherence to PAP duration: Adherence to appropriate perioperative antibiotic prophylaxis (PAP) duration significantly improved after the intervention but remained low overall. This suggests the intervention had some effect, but more efforts might be necessary to improve adherence rates.

4: Factors Associated with Adherence to Appropriate Duration: Variables associated with adherence to PAP duration included the phase of the study (POST), preoperative hospitalization of 3 days or more, duration of operation of 4 hours or less, and the identity of the chief surgeon. This suggests that both organizational (phase of the study, hospital stay) and procedural (operation duration, surgeon) factors can influence adherence to PAP duration.

Need to be polished and improved

Author Response

We thank the reviewer for the comments. Please find attached the detailed response.

Reviewer 2 Report

The manuscript entitled “Effect of an e-prescription intervention on the adherence with 2 surgical chemoprophylaxis duration in cardiac surgery” is presenting a very important issue of how PAP can control post-operative infection development. I evaluate this work as a screening one, just "a single-center experience" . It has educational value, but In my opinion, the article is not entirely written and should be improved.

All comments are in the attached file.

Author Response

We would like to thank the reviewer for the useful comments. Please find attached the response.

Round 2

Reviewer 1 Report

The authors have addressed the concerns. I do not have any further comments.

Reviewer 2 Report

 The Authors present the original paper. The title is consistent with the problem presented and contains everything to guess about the article. The abstract reflects the content of the article. The most important information is included and condensed in the abstract. The conclusion is clear. Point-to-point answers are satisfactory. 

All of this is required for the further evaluation process.